# Improving EEG-Based Driver Distraction Classification Using Brain Connectivity Estimators

**DOI:** 10.3390/s22166230

**Published:** 2022-08-19

**Authors:** Dulan Perera, Yu-Kai Wang, Chin-Teng Lin, Hung Nguyen, Rifai Chai

**Affiliations:** 1School of Science, Computing and Engineering Technologies, Swinburne University of Technology, Hawthorn, VIC 3122, Australia; 2School of Computer Science, University of Technology Sydney, Ultimo, NSW 2007, Australia

**Keywords:** distracted driving, brain connectivity, GGC, DTF, PDC, GPDC, driver distraction classification, PSD, SVM

## Abstract

This paper discusses a novel approach to an EEG (electroencephalogram)-based driver distraction classification by using brain connectivity estimators as features. Ten healthy volunteers with more than one year of driving experience and an average age of 24.3 participated in a virtual reality environment with two conditions, a simple math problem-solving task and a lane-keeping task to mimic the distracted driving task and a non-distracted driving task, respectively. Independent component analysis (ICA) was conducted on the selected epochs of six selected components relevant to the frontal, central, parietal, occipital, left motor, and right motor areas. Granger–Geweke causality (GGC), directed transfer function (DTF), partial directed coherence (PDC), and generalized partial directed coherence (GPDC) brain connectivity estimators were used to calculate the connectivity matrixes. These connectivity matrixes were used as features to train the support vector machine (SVM) with the radial basis function (RBF) and classify the distracted and non-distracted driving tasks. GGC, DTF, PDC, and GPDC connectivity estimators yielded the classification accuracies of 82.27%, 70.02%, 86.19%, and 80.95%, respectively. Further analysis of the PDC connectivity estimator was conducted to determine the best window to differentiate between the distracted and non-distracted driving tasks. This study suggests that the PDC connectivity estimator can yield better classification accuracy for driver distractions.

## 1. Introduction

The driver must keep full attention to control the vehicle as the driving task requires the driver’s full attention [1]. Statistics given by the World Health Organization indicate that 1.3 million people die per year due to roadside accidents worldwide. In the present day, driver distractions have become a huge concern among commuters on the road [2,3,4]. A study found that 6.7% of middle-aged drivers and 8.8% of elderly drivers engage in distracting activities that could lead to a high risk of accidents [1]. Distractions can be classified as the devices or activities that lead the driver’s attention away from the driving task [5]. Primarily, driver distractions can be classified into four main categories. (i) Auditory distraction: listening to something unrelated to driving while driving is considered an auditory distraction [6]; (ii) visual distraction: glancing at something other than the road while driving is considered a visual distraction [5,7]; (iii) cognitive distraction: thinking about things unrelated to the driving task while driving is classified as a cognitive distraction [5]; and (iv) manual distraction: participating in activities unrelated to driving while driving is classified as a manual distraction [5]. These above-mentioned categories can interact together to create distractions. The aforementioned distractions can result in injuries, property damage, and sometimes fatalities. Numerous efforts have been taken to detect driver distractions promptly to develop a reliable system to support the drivers accordingly [8,9].

Prominent techniques to detect driver distractions are monitoring the driver’s behavior using a camera and image processing technique or monitoring the brain’s activity using an electroencephalogram (EEG) [10,11,12,13,14,15]. When monitoring the driver’s behavior, image processing techniques are used to detect either the eye movement or the driver’s activities using a camera. Physiological assessment of facial or eye movements using captured images or video recordings of the driver’s face may lead to privacy issues compared to the physiological measurement methods [16]. Furthermore, an EEG [17,18] directly measures the neurophysiological signals from the source, and it can easily be correlated with distractions [19,20,21,22], driver fatigue [13,23], and drowsiness [24].

When designing an EEG-based classification countermeasure system, EEG signal measurement and preprocessing, feature extraction, and classification modules are essential. During the EEG signal measurement and preprocessing phase, data acquisition and the initial preprocessing are conducted. The popular feature extraction method in brain monitoring is based on frequency analysis such as power spectral density and fast Fourier transform. Different EEG frequency bands were also used in mental fatigue classification [25]. An automatic EEG classification of EEG for dementia stages was investigated by using wavelet analysis to construct five EEG bands [26]. In event-related desynchronization/synchronization (ERD/ERS)-based BCI, it used mu rhythm (9–13 Hz) as a feature [27]. Power spectral density shows the strength of each frequency [28]. To obtain more useful features, it is recommended to consider the relationships between EEG source/sensor signals in brain connectivity [29], in which brain connectivity estimators can show the relations between each selected brain area. Thus, this paper proposes to use the brain connectivity method as a feature extractor in the EEG-based classification of distracted and non-distracted driving tasks.

Brain connectivity estimators can describe the organization of the brain and patterns of links. Brain connectivity can be divided into three main categories. (i) Structural connectivity [30], where anatomical connections are described. (ii) Functional brain connectivity [31,32], where statistical dependence patterns are captured. (iii) Effective brain connectivity [33] describes the influence of one neural system over another.

Functional brain connectivity can be further divided into two subcategories: time domain functional brain connectivity and frequency domain functional brain connectivity [34]. One of the most popular time domain functional brain connectivity estimator methods is the Granger–Geweke causality (GGC) connectivity estimation method [35], whereas for the frequency domain directed transfer function (DTF) [36], partial directed coherence (PDC) [37,38], and generalized partial directed coherence (GPDC) [39] are some of the most commonly used brain connectivity estimators [40]. In our previous study [41], we were able to conclude by using the Student *t*-test and the Anova test that there is a difference between the connectivity values of the distracted driving and non-distracted driving tasks for the GGC, PDC, and DTF connectivity estimators. In this study, we were able to conclude that the PDC has the highest classification accuracy. GPDC is a modified variant of PDC.

When the classification modules are considered, artificial intelligence (AI) plays an important role. Artificial intelligence (AI) is a method to mimic human decision-making procedures. Machine learning is an AI type where the software application is capable of making accurate predictions without being told to [42]. Prominent machine learning subcategories are supervised and unsupervised learning [43]. Due to being one of the better performing linear classifiers and having low computational complexity, the support vector machine (SVM) is one of the popular supervised learning models. SVM can be used to develop a classification prediction model using input and output data [17,44]. Support vector machine classifications can be subcategorized into two main categories, such as binary classification and multiclass classification [45], where binary support vector machines can be further categorized as binary linear classification and binary kernel classification [46].

The main contribution of this paper is the novel approach of using brain connectivity [29] estimators as features to classify distracted driving and non-distracted driving tasks, which have not been explored previously for the driver distraction classification to improve the classification accuracy. This paper will investigate a few connectivity analysis estimators as features for classification.

The structure of the paper is as follows: Section 2, materials and the methods, covers the general structure of the experiment, data collection, experiment conditions, EEG data processing, connectivity analysis, and classification. In Section 3 are the results of the independent component analysis, connectivity analysis results, and classification accuracy results. Section 4 discusses the results, and Section 5 follows up with the study’s conclusion.

## 2. Materials and Methods

### 2.1. General Structure

The general structure of the study is shown in Figure 1. In the initial stage, driver distraction data were collected from 10 participants. Data were collected using 32 EEG channels and four 15 min sessions per participant [23]. In the next stage of the study, collected data were filtered and downsampled before the necessary event-related epoch extractions. Next, an independent component analysis (ICA) was conducted on the necessary data. Furthermore, relevant components were selected, and the connectivity analysis was conducted for each required condition in the EEG band from 1 Hz to 20 Hz. In the final stage, both distracted and non-distracted driver data were divided into a test set and a training set at a ratio of 50:50. Then support vector machine (SVM) model with radial basis function (RBF) kernel was trained using the training data to test the classification of the testing data.

### 2.2. Data Collection

The experiment for the data collection was conducted using 10 healthy participants. The average age of the participants was 24.3 (SD 2.05), with a minimum driving experience of 1 year. Furthermore, male to female ratio was 9:1 for this study, and the data were collected from 1 pm to 4 pm on a given day. The study was conducted with the recommendations and the accordance of Taipei Veterans General Hospital. Taipei Veterans General Hospital approved the protocol for the study, and all subjects gave written consent. All the participants had a normal or corrected vision and were forbidden to take any drugs, caffeine, or alcohol before the experiment. Two sessions of 15 min were used as training sessions for each participant to become familiar with the environment, the lane-keeping task (non-distracted driver data), and the problem-solving task (distracted driver data).

A dynamic motion simulator with a simulation environment was used to obtain more realistic data. The motion simulator was a real car with a 3D simulated environment mounted on a 6-DOF motion platform. As shown in Figure 2 and Figure 3, simulation scenes were developed using World Tool Kit (WTK) library. Six screens were used, with the frontal field of view of 206° and the backfield of view of 40°. The size of each screen had a diagonal measuring of 2.6 m–3.75 m. In the simulation environment, the car was cruising at a speed of 100 km/h in the third lane of a four-lane highway. Car speed being fixed at 100 km/h is a limitation in this study. Two experimental conditions were introduced randomly throughout the sessions to collect distracted data and non-distracted driver data. A lane-keeping task was introduced to collect the non-distracted driver data, whereas a math problem-solving task was introduced to collect the distracted driver data.

Non-distracted driver data were collected by using the car’s condition gradually drifting randomly towards the right or the left side of the designated lane (lane 3). The participant had to move the car back to the designated lane. To collect the distracting data, a simple math equation appeared on the screen. The participant had to confirm whether the equation is correct or incorrect by pressing the designated buttons on the steering wheel. Correct to incorrect equation appearance rate was 50:50, and the complexity of the equation remained the same throughout the experiment. The right-side button of the steering wheel was allocated for the correct answers, and the left-side button on the steering wheel was assigned for the incorrect answers. The 6-s to 8-s intervals were introduced between the two tasks [21]. In this study, we selected only the data with correct responses.

### 2.3. EEG Data Acquisition and Preprocessing

A modified 10/20 BCI system with 32 Ag/AgCl EEG channels with the NuAmps Express system was used for the EEG data acquisition. A 16-bit quantization at a frequency of 500 Hz was used for the data collection. The use of the conductive gel helps the impedance to be under 10 Kohm. Channel locations and the raw EEG data sample are shown in Figure 4. The 32 channels used in this study include *FP1*, *FP2*, *F8*, *F4*, *Fz*, *F3*, *F7*, *FC4*, *FCz*, *FC3*, *C4*, *Cz*, *C3*, *CP4*, *CPz*, *CP3*, *P4*, *Pz*, *P3*, *T6*, *T5*, *T8*, *O2*, *Oz*, *O1*, *F8*, *F7*, *TP8*, *TP7*, *A1,* and *A2*.

MATLAB’s EEGLAB extension was used for the preprocessing of the data. All the EEG data collected from the participants were downsampled to 250 Hz. A 0.5 Hz high pass filter was used to remove the DC drift and the noise. A 50 Hz low pass filter was used. To obtain the 0.5 Hz low pass filter and the 50 Hz high pass filter, the pop_eegfiltnew() function with the input parameters of lower edge 0.5, the higher edge of 50, and second order filter was used. Relevant EEG data for the Math problem-solving task and the Lane keeping task were extracted from the continuous EEG data. Reference *A1* and *A2* were removed before the analysis. Furthermore, channels *FP1* and *FP2* were removed to negate the effect of blinking. Figure 4 shows the used channels in this study.

### 2.4. Preprocessing: Independent Component Analysis

Independent component analysis (ICA) was conducted to remove the artifact further and select the required component for the brain regions [48]. Independent components can be determined as follows
(1)S=WX  
where S is the source activity, W is the weight matrix, and X is data in the original space. EEGLAB’s RUNICA plugin was used to decompose data using the Infomax-ICA algorithm [49]. To cover the activities from the frontal, central, parietal, occipital, left, and right motor areas, in this study, brain components covering frontal, central, parietal, occipital, left motor, and right motor were selected. EEGLAB’s independent component label plugin was used first to classify the components. After that, with the help of an expert in ICA, relevant brain components were selected. ICA component label plugin failed to classify a proper right motor component. Hence, participant 7 was removed from the study analysis.

### 2.5. Feature Extractions: Power Spectral Density Analysis

After removing the noise and artifacts, EEG signals of the selected six components were used to estimate the power spectral density of the signal [28]. Spectral density was calculated by using the bias estimation of the autocorrelation sequence; in other words, the periodogram. The following equation can be used to determine the periodogram.
(2)$$\hat{p}\left(f\right)=\frac{\Delta t}{N}{\left|\sum_{n=0}^{N-1}x_{n}e^{-j2\pi f\Delta tn}\right|}^{2};-\frac{1}{2}\Delta t<f\leq \frac{1}{2}\Delta t\;$$
where single $x_{n}$ is sampled at f at a unit time and Δt is the sampling interval. EEG data containing the six components were used separately to calculate the power spectral density using the periodogram function. This yields 1542 features for each epoch of distracted and non-distracted driving. The data set was divided into training and testing data sets with a ratio of 50:50 before the classification steps.

### 2.6. Feature Extractions: Brain Connectivity Analysis Structure

Brain connectivity analysis can be divided into three main categories. First, the model order selection, then the multivariate autoregressive model (MVAR), and finally, the connectivity estimation.

Connectivity estimation heavily depends on the MVAR model reliability. Model order and the epoch length have a considerable effect in MVAR modeling [50]. Non-distracted epochs in this study have a length of 1200 ms, and for distracted, it is 1600 ms. In MVAR modeling, it is crucial to select suitable sliding windows as it will not lose any data while processing. Furthermore, using higher or lower time duration can cause the connectivity analysis to be redundant [51]. To calculate the optimal MVAR model, EEG epoch was divided into steps with a length of 400 ms and an overlap of 50 ms. Overlap time windows make the estimation model smooth [52]. Furthermore, longer time steps will lose the temporal dynamics [52].

To calculate the MVAR model, model order should be calculated. To estimate the optimal model order EEGLAB’s Source Information Toolbox (SIFT) was used [53]. Bayesian information criterion (BIC) (Schwarz–Bayes criterion (SBC)), Akaike’s information criterion (AIC), Hannan–Quinn criterion (HQ), and the Akaike’s final prediction error criterion (FPE) with the elbow of the mean curve and the min of mean curve methods were used. Model order range from 1 to 30 was selected to estimate the optimal model order for the given epoch.

The multivariate autoregressive model is the base of brain connectivity estimators such as Granger–Geweke causality (GGC), directed transfer function (DTF), partial directed coherence (PDC), generalized partial directed coherence (GPDC). The following equation can be used for the AR model interpretation.
(3)$$X\left(t\right)=\sum_{j=1}^{p}A\left(j\right)X\left(t-j\right)+E\left(t\right)\;$$
where sample data X(t) is given by the sum of previous *p* samples from the set of k signal weighted model coefficient *A* and random *E* value, where *p* is the model order. Model order *p* can be estimated by using the Bayesian information criterion (BIC) (Schwarz–Bayes criterion (SBC)), Akaike’s information criterion (AIC), Hannan–Quinn criterion (HQ), and the Akaike’s final prediction error criterion (FPE).

Granger–Geweke causality index was used as a time domain connectivity estimator. GGC index (GCI) can be calculated by using the following equation.
(4)$${\mathrm{GCI}}_{i\to j}\left(t\right)=\ln\left(\frac{V_{i,n}\left(t\right)}{V_{i,n-1}\left(t\right)}\right)\;$$
where $V_{i,n}\left(t\right),\;V_{i,n-1}\left(t\right)$ denotes the residual variance for *n*, *n* − 1 dimensional MVAR; i and *j* are the channels by which GCI is calculated.

The directed transfer function (DTF) is a frequency domain brain connectivity estimator. DTF connectivity values can be estimated by using the function described in (3). This function explains the influence of channel *j* on channel *i*.
(5)$$\gamma_{ij}^{2}\left(f\right)=\frac{{\left|H_{ij}\left(f\right)\right|}^{2}}{\sum_{m=1}^{k}{\left|H_{im}\left(f\right)\right|}^{2}\;}\;$$
where elements of the multivariate autoregression model transfer function matrix are denoted as $H_{ij}\left(f\right)$.

Partial directed coherence (PDC) can be used for the detection of the directed and cascade flows [54]. PDC function describes the influence of channel *i* on channel *j*. PDC can be determined as follows
(6)$$P_{ij}\left(f\right)=\frac{A_{ij}\left(f\right)}{\sqrt{a_{j}^{*}\left(f\right)a_{j}\left(f\right)}}\;$$
where $A_{ij}\left(f\right)$ denotes an element from the Fourier transform matrix from the multivariate autoregression model coefficient A(t) and *j*th column of A(f) is denoted as $a_{j}$.
(7)$$g{\mathrm{PDC}}_{ij}\left(f\right)=\left|\frac{\frac{1}{\sigma }A_{ij}\left(f\right)}{\sqrt{\sum_{i=1}^{m}\frac{1}{\sigma_{i}^{2}}{\left|A_{ij}\left(f\right)\right|}^{2}}}\right|{\;}^{2}\;$$

In this equation $A_{ij}\left(f\right)$ yields an element from the Fourier transform matrix from the multivariate autoregression model coefficient A(t). Furthermore, $\sigma_{i}^{2}$ is the residual variance of the variable *i*.

### 2.7. Classification and Optimization

Each connectivity estimator matrix was divided into training and testing data sets with a ratio of 50:50. Source, target, frequency, and MVAR model time steps were used as the classification features. Furthermore, in the power spectral density analysis, frequency features of distracted and non-distracted driving were divided into training and testing data sets with a ratio of 50:50.

Radial basis function (RBF) SVM or Gaussian SVM can be determined as follows.
(8)$$K\left(x_{1},x_{2}\right)=exp\left(\frac{\|x_{1}-x_{2}\|{}^{2}}{2\sigma^{2}}\right)$$
where $x_{1},x_{2}$ denotes the data points, and σ indicates the width of the kernel. Bayesian optimization was used with the expected improvement plus acquisition function to optimize the training model. Furthermore, 30 object evaluations were considered in the optimization process [55]. Hyperparameters were tuned by minimizing the five-fold cross-validation loss using the Bayesian optimizer and expected improvement plus acquisition function.

## 3. Results

### 3.1. Independent Component Analysis

Independent component analysis (ICA) was used on both extracted epochs of distracted and non-distracted driving. Twenty-eight independent components were formed from the twenty-eight channels. Figure 5 shows the non-distracted driving output of the ICA for a participant, whereas Figure 6 shows the distracted driving output of the ICA for the same participant.

EEGLAB’s ICLabel tool was first used to determine the components to remove the noisy components. EEGLAB’s ICLabel tool initially classified the formed components as brain, muscle, eye, and other components. The output of a participant’s independent component label toolbox is shown in Figure 7 and Figure 8. It shows the filter output for the same participant’s non-distracted and distracted driving tasks shown in Figure 5 and Figure 6, respectively.

Brain components were selected from the ICA analysis, and the noise components were removed. A previous study [47] showed that the frontal, central, parietal, left motor and right motor areas are more useful in driver distraction detection. Hence, with the expert’s help, relevant components for the frontal, central, parietal, left motor, and right motor were selected. Selected independent components of non-distracted and distracted tasks for a participant are shown in Figure 9 and Figure 10. ICs 2, 3, 4, 8, 12, and 17 in Figure 6 and Figure 8 are equivalent to ICs 1, 2, 3, 4, 5, and 6 in Figure 10, respectively. ICs 2, 3, 6, 7, 10, and 18 in Figure 5 and Figure 7 are equivalent to ICs 1, 2, 3, 4, 5, and 6 in Figure 9, respectively.

### 3.2. Model Order Calculation

SIFT was used to calculate the model order. Figure 11 shows the elbow of the mean curve plot for participant 1. All information criteria (SBC, AIC, FPE, HQ) yield the model of 5 when the elbow of the mean curve method is selected.

The mean curve minimum method with SBC, AIC, FPE, and HQ information criteria was used to find the optimal model order for each participant’s distracted and non-distracted driving tasks. Table 1 shows the summary of model order selection for the distracted and non-distracted driving tasks for all the participants. For both scenarios, the SBC elbow of the mean curve and the min of the mean curve yielded the model order of five. In comparison, the AIC elbow of the mean curve for the distracted driving and non-distracted driving task yielded the model order of five, and the mean curve minimum for both scenarios yielded the model order of nine. For both the distracted driving scenario and the non-distracted driving scenario, the FPE elbow of the mean curve gave the model order of five, and the min of the mean curve yielded the model order of nine for both scenarios. For both the distracted and non-distracted driving scenarios, the HQ elbow of the mean curve yielded the model order of five, and the minimum of the mean curve yielded eight. In this study, for the multivariate auto-aggressive model calculation model order, five was used.

After the model order was determined, the Granger–Geweke causality connectivity estimator, partial directed coherence connectivity estimator, generalized partial directed coherence connectivity estimator, and directed transfer function connectivity estimator connectivity matrix were used to calculate the connectivity matrixes for each epoch. The final connectivity matrix for each estimator has the dimensions of 6 × 6 × 20 × 17 for non-distracted driving and 6 × 6 × 20 × 25 for distracted driving, whereas it represents the source × target × frequency × time steps. Source and target represent the six components, frontal, central, parietal, occipital, right motor, and left motor. This study considered frequencies from 1 Hz to 20 Hz for the binary classification between the distracted and non-distracted driving tasks. 

### 3.3. Classification

MATLAB’s fitcsvm function with the RBF kernel was used to train and classify the model. The inputs for this function are EEG connectivity features and SVM parameters, which require optimization. Output from this function yields the optimized SVM model. Data preparation for the SVM function is as follows, distracted and non-distracted epochs were separated for each subject, and for each subject, epochs were divided in the ratio of 50:50 as the training and testing data set. The training data were mixed in order from different participants before feeding them into the SVM training function. Furthermore, 30 object evaluations with Bayesian optimization were used to find the optimized training model. Figure 12 shows the minimum objective for each function evaluation from 1 to 30 function evaluations for the PDC connectivity estimator. Each evaluation minimizes the five-fold cross-validation loss by tuning the hyperparameters automatically.

The object function model for the PDC connectivity estimator is shown in Figure 12. Furthermore, box values and sigma values for each estimated object value are shown in Figure 13. The selected optimized box value and the sigma value are used as the values for the parameters for box constraint and the kernel scale, respectively.

After the trained model was optimized for each feature acquisition method, the testing data set was used to calculate the classification accuracy of each relevant SVM model. Classification accuracy results for the testing data set with the optimal SVM model are shown in Table 2.

DTF connectivity estimator yielded a 70.02% classification accuracy for distracted and non-distracted driving. Whereas GGC yielded 82.17%, PDC yielded 86.19%, and GPDC yielded 80.95%. Furthermore, the features using the conventional method of EEG data analysis, in other words, power spectral density (PSD), yielded a classification accuracy of 74.05%.

The highest classification accuracy was obtained using the PDC connectivity estimator. Hence, the PDC connectivity estimator was selected for further analysis. To determine the best time window to separate the distracted and non-distracted driving tasks, features containing connectivity matrixes for each time window were used to train and optimize an SVM model with an RBF kernel. Figure 14 shows the classification accuracies for the 17 time windows.

## 4. Discussion

This study aims to investigate and compare the brain connectivity estimators as the features to classify distracted vs. non-distracted driving. To mimic distracted driving, a math problem-solving task was introduced, and to mimic the driving task, a lane-keeping task was introduced in the experiment. Directed transfer function (DTF), Granger–Geweke causality (GGC), partial directed coherence (PDC), and generalized partial directed coherence (GPDC) connectivity estimators were considered in our study.

Connectivity matrixes for the distracted and non-distracted driving tasks containing the frontal, parietal, central, left motor, and right motor occipital independent components were estimated. After that, the SVM with an RBF kernel was used for the classification of the distracted and non-distracted driving tasks. The highest classification accuracy obtained when using the PDC brain connectivity was 86.19%. Furthermore, features obtained through GGC and GPDC connectivity estimators yielded a classification accuracy of 82.27% and 80.95%, respectively. Moreover, features obtained by the conventional method of EEG data analysis, power spectral density (PSD), yielded the classification accuracy of 74.05%. With the above results, it is safe to assume that the features obtained using PDC, GGC, and GPDC connectivity estimators have a better classification accuracy than the features obtained through the power spectral density analysis for this given data set. However, features obtained through the DTF connectivity estimator have a lower classification accuracy than those of the PSD. Table 2 shows the classification accuracy summary for each type of feature. As shown in Figure 14, the highest classification accuracy of 73.19% was obtained between distracted and non-distracted driving during the 200 ms of the onset stimulus and 600 ms of the onset stimulus, whereas the lowest accuracy of 62.62% was obtained during the 700 ms and 1100 ms window step. Hence, we can safely assume that 200 ms to 600 ms of the stimulus is the best time to classify the distracted and non-distracted driving tasks for this given data set.

From the above conclusions, the time window of 200 ms to 600 ms was further analyzed. Connectivity values between each component were compared between the distracted and non-distracted driving tasks. Connectivity estimations between distracted and non-distracted driving are shown in Figure 15 and Figure 16. For visualization purposes, an average of nine participants of the connectivity matrixes was taken. Then the average was displayed for a given time window in Figure 15 and Figure 16 to have a general idea of the brain connectivity throughout the distracted and non-distracted driving scenarios. Figure 15 shows the important brain connections obtained by using the PDC connectivity estimator for the distracted driving of a participant, where IC1 is the central component, IC2 is the parietal component, IC3 is the occipital component, and IC4 is the frontal component. Connectivity between the parietal and occipital components during distracted driving is the highest for the given data set, whereas connectivity between the central and frontal areas is the second highest. Furthermore, connectivity power between the central and occipital areas is the third highest, and connectivity power between the parietal and frontal areas is the fourth.

The non-distracted driving task connectivity brain component map for a participant is shown in Figure 16. IC components are similar to the above-mentioned components. During the distracted driving scenario, high connectivity can be seen between the central and occipital components. Moderate connectivity power can be seen between the central and frontal components compared to the distracted driving scenario at the given time window of 200 ms to 600 ms of the onset stimulus while using the PDC connectivity estimator for the selected participant.

In this study, brain connectivity was used to investigate driver distractions. The above results indicate how the brain networks become dense when the driver is distracted. When the driver is distracted, brain connections between multiple regions become significant. Moreover, compared to the non-distracted driving scenario, distracted driving causes more correlations between numerous regions of the brain. Hence, the use of brain connectivity estimators to differentiate between distracted and non-distracted driving is an effective method. Furthermore, the partial directed coherence brain connectivity estimator yields a better classification accuracy than the conventional method of power spectral analysis.

## 5. Conclusions

In this study, ten participants who participated in a simulated driving experiment with two experimental conditions were analyzed. A math problem-solving task was considered a distracted driving scenario, and the lane-keeping task was considered a non-distracted driving scenario. Brain connectivity estimators were used as the features for the SVM classifier. The highest accuracy of 86.19% was obtained when using the PDC connectivity estimators as the features. Moreover, this study compared different brain connectivity methods and the conventional EEG features based on PSD.

For the application in driver distraction detection methods, this paper offers a unique insight into using brain connectivity estimators as features for the classification. Results of this study suggest that to detect driver distractions the partial directed coherence (PDC) connectivity estimator is better suited as features compared to the direct transfer function (DTF), Granger–Geweke causality (GGC), and generalised partial directed coherence (GPDC) connectivity estimators.

The main challenges in detecting driver distractions in real time would be choosing the optimal method to acquire features and using the optimal number of EEG channels for the detection. In this paper, the optimal method for driver distraction is proposed, and selecting the optimal number of EEG channels remains challenging. These results are expected to provide a foundation to develop driver distraction detection methods in a real-time environment to reduce the roadside fatalities caused by distracted drivers.

## Figures and Tables

**Figure 1 sensors-22-06230-f001:**
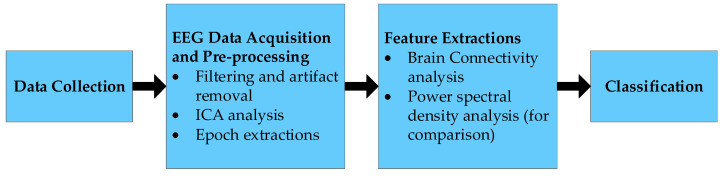
Graphical description of the analysis for the study.

**Figure 2 sensors-22-06230-f002:**
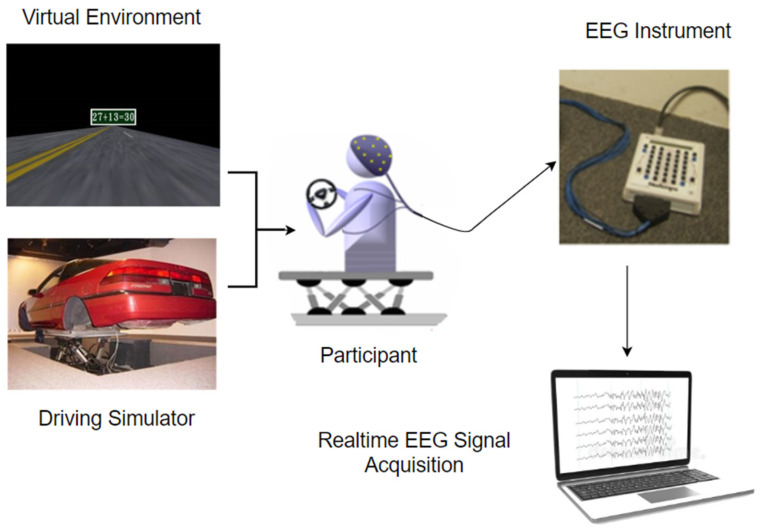
Graphical overview of the experiment data collection process.

**Figure 3 sensors-22-06230-f003:**
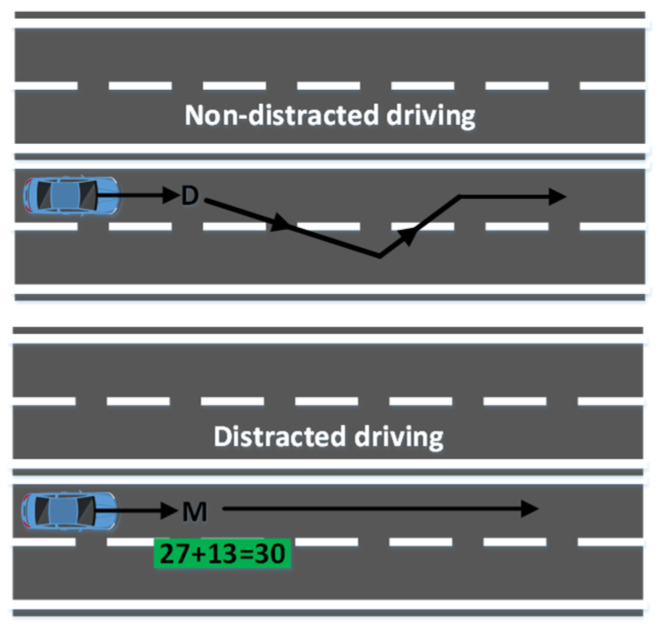
Two experimental conditions D: lane deviation occurs and M: math problem appears [47].

**Figure 4 sensors-22-06230-f004:**
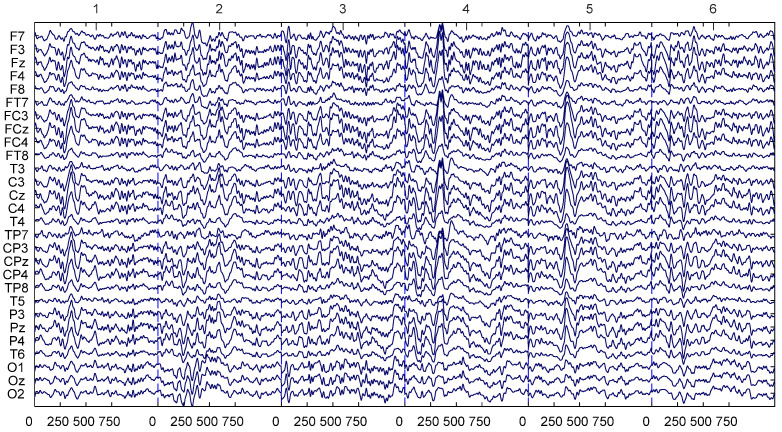
EEG signal for the channels used in this study.

**Figure 5 sensors-22-06230-f005:**
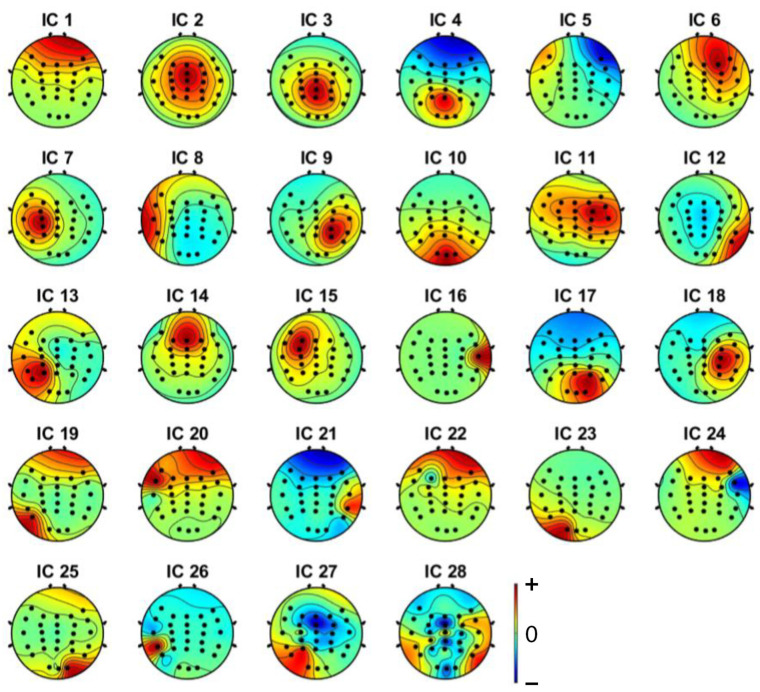
Independent components (from IC1 to IC28) for non-distracted driving tasks of a participant.

**Figure 6 sensors-22-06230-f006:**
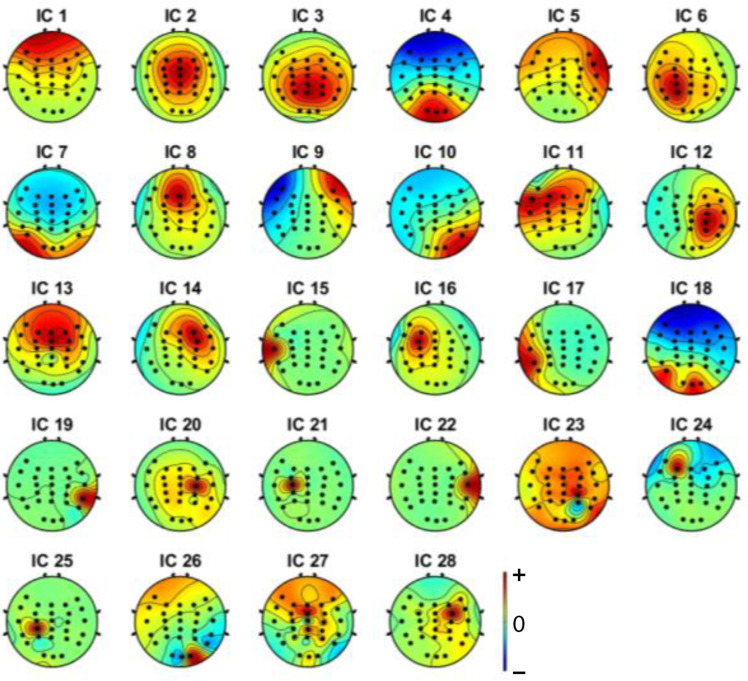
Independent components (from IC1 to IC28) for distracted driving tasks of a participant.

**Figure 7 sensors-22-06230-f007:**
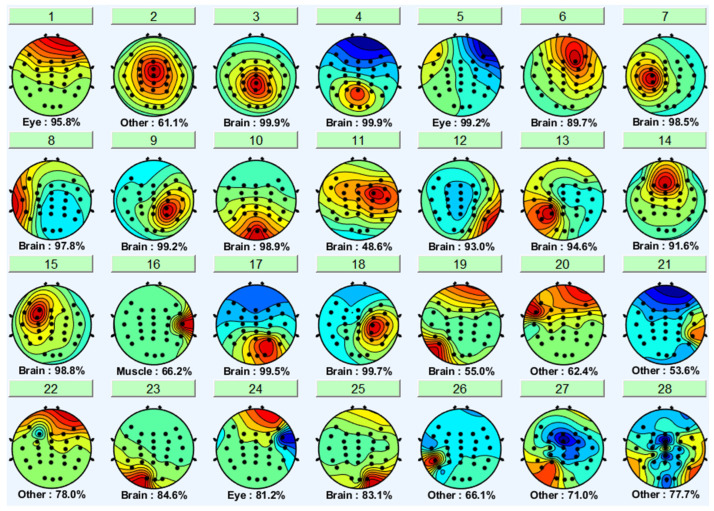
The output of EEGLAB-ICA label function (brain signal vs. noises) from non-distracted driving.

**Figure 8 sensors-22-06230-f008:**
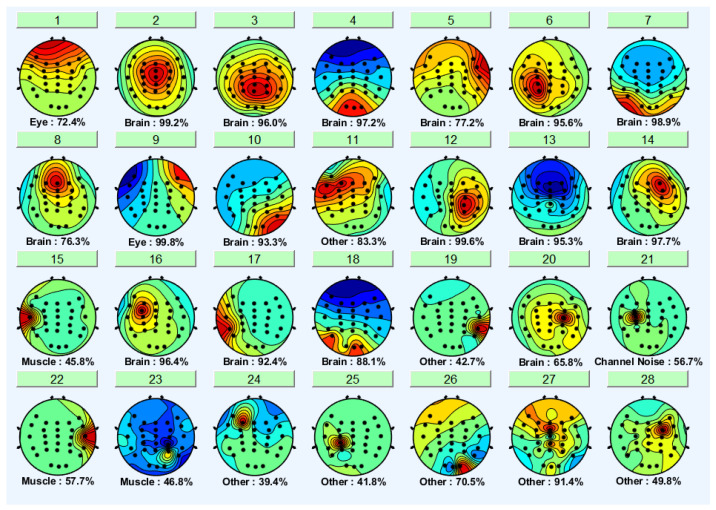
The output of EEGLAB-ICA label function (brain signal vs. noises) from the distracted driving task.

**Figure 9 sensors-22-06230-f009:**
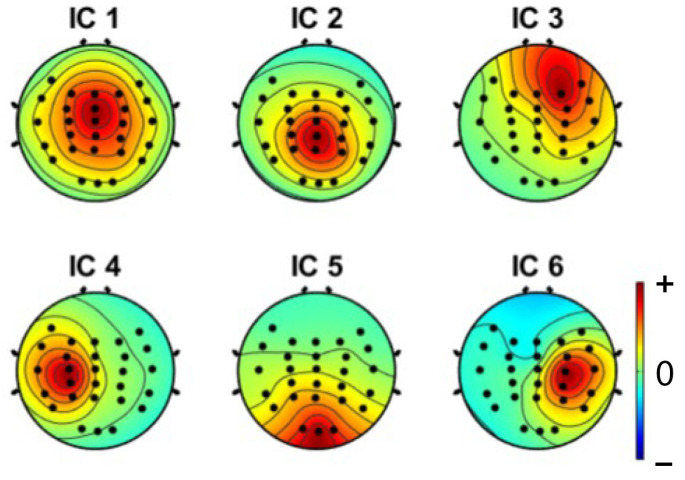
Selected independent components of a participant, where IC1-central, IC2-parietal, IC3-frontal, IC4-left motor, IC5-occipital, and IC6-right motor components are shown.

**Figure 10 sensors-22-06230-f010:**
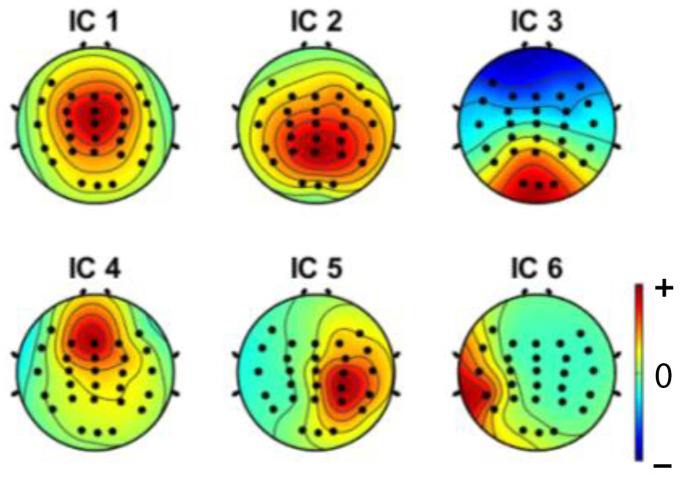
Selected independent components of a participant, where IC1-central, IC2-parietal, IC3-occipital, IC4-frontal, IC5-right motor, and IC6-left motor components are shown.

**Figure 11 sensors-22-06230-f011:**
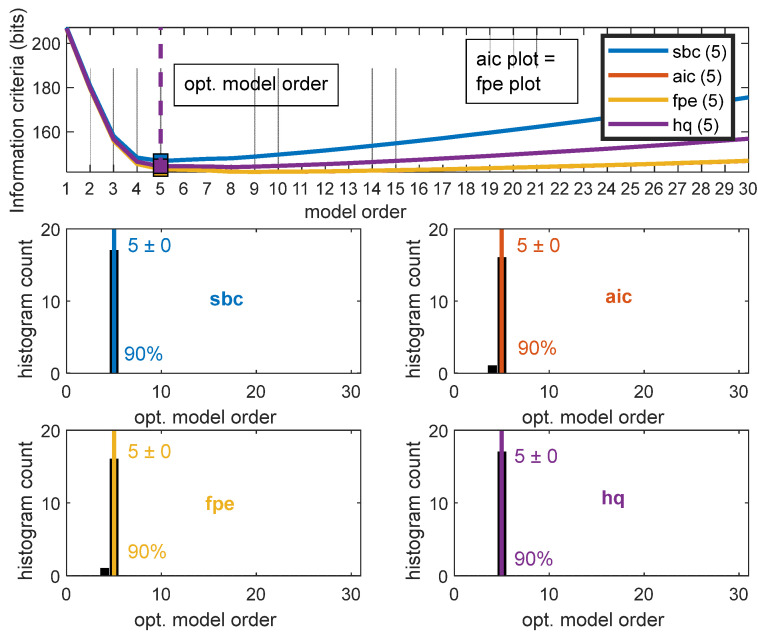
Model order selection elbow of the mean curve method for a participant.

**Figure 12 sensors-22-06230-f012:**
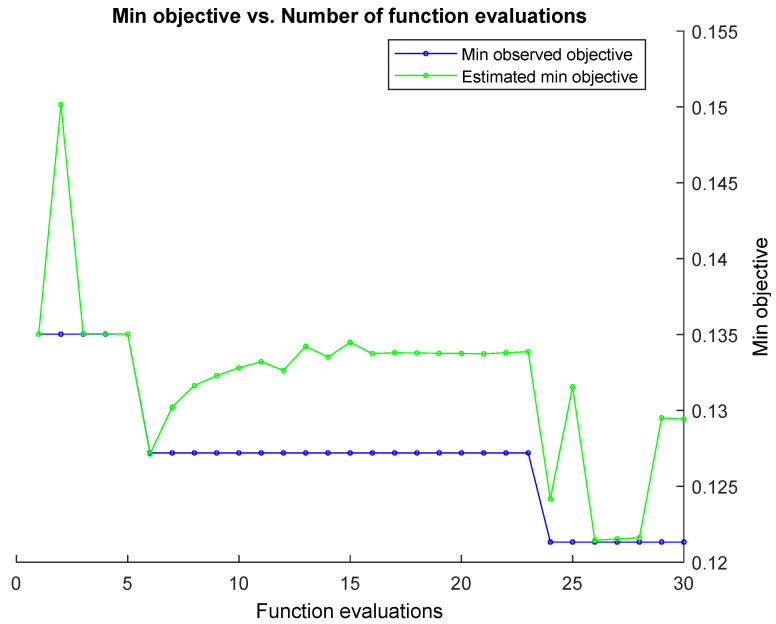
Support vector machine optimization function evaluations vs. min objective for GGC brain connectivity estimator.

**Figure 13 sensors-22-06230-f013:**
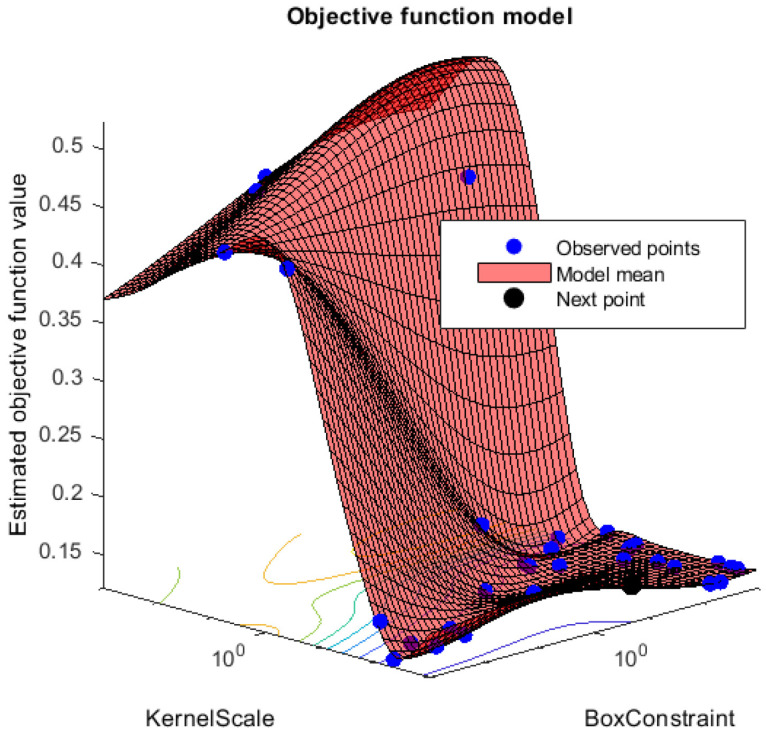
Support vector machine optimization, optimized box constraint value, and the sigma value for GGC brain connectivity estimator.

**Figure 14 sensors-22-06230-f014:**
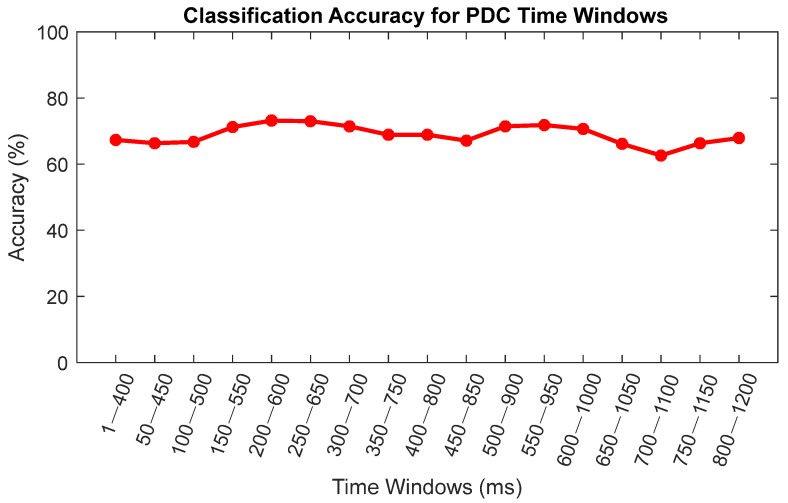
Classification accuracies for PDC time windows.

**Figure 15 sensors-22-06230-f015:**
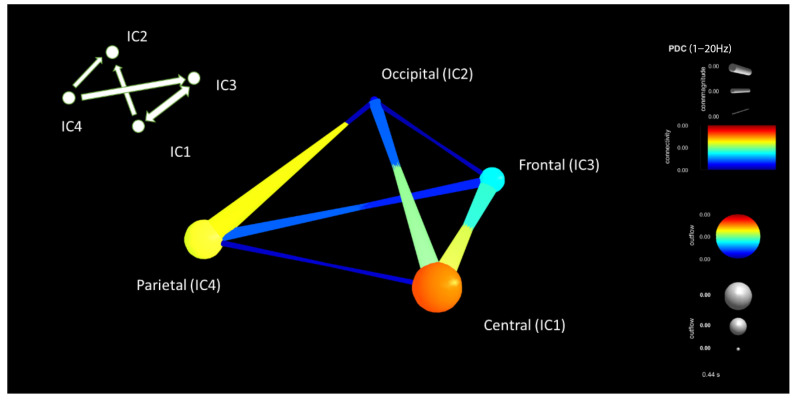
Brain connectivity visualization for the period window of 200–600 ms for distracted driving for the PDC connectivity estimator.

**Figure 16 sensors-22-06230-f016:**
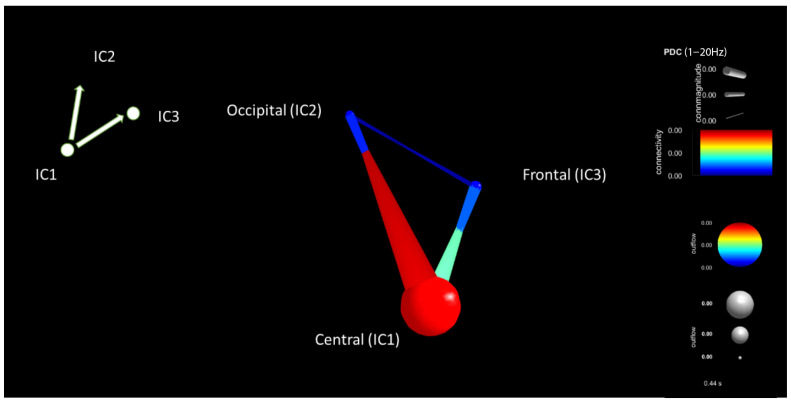
Brain connectivity visualization for the period window of 200 ms–600 ms for non-distracted driving for the PDC connectivity estimator.

**Table 1 sensors-22-06230-t001:** Model order selection summary.

Criteria	Distracted Driving		Non-Distracted Driving	
	Elbow	Min	Elbow	Min
SBC	5	5	5	5
AIC	5	9	5	9
FPE	5	9	5	9
HQ	5	8	5	8

**Table 2 sensors-22-06230-t002:** Classification accuracy summary.

Features for the Classification	Classification Accuracy
Power Spectral Analysis	74.05%
DTF	70.02%
GGC	82.27%
PDC	86.19%
GPDC	80.95%

## Data Availability

Third party data were obtained from Yu-Kai Wang and are available with the permission of National Chiao Tung University.

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
