# Peer review of "Improving EEG-Based Driver Distraction Classification Using Brain Connectivity Estimators"

_sensors, 2022, doi:10.3390/s22166230_

Round 1

Reviewer 1 Report

Dear authors

thanks for your work. I have made below suggestions and comments on your work that is worthy of publication but would require major changes in my opinion to make it clearer how the study was conducted and the data analyzed.

Reference 1 refers to a really specific clientele of drivers that is clearly not the one reference here. I suggest the authors change it for other works either from Reagan and Lee or Hancock on the topic of the general driver. Same goes for reference #3.

Line 44: Thi should also be explained that these 4 categories can also interact all together.

Ref 14 is an EEG based study and it is used to highlight privacy issue for image based recognition, I would suggest using a more appropriate reference on the topic.

Line 62 remains unclear. The statement of research : "analyze the data" of what ? It has to be clearer.

The paragraph in line 76 needs a proper sequence from the previous manuscript. It is well written but as no connection with the paragraph either above or after it.

There is an unnecessary paragraph at line 90.

Line 101 to 106 are unnecessary in my opinion since the manuscript follows a typical scientific paper.

Section 2.2: The method has to report the SD of age and sex/gender ratio of participants. Moreover, the time at which the data collection was done would be important for reproducibility. Moreover, years of experience is also necessary.

The simulator has to be better described : screen size, controls, software used, etc. Same apply to the EEG system where the company and model used should be presented.

A rational for the down-sampling has to be given going from 500 to 250 Hz.

Figure 4 and 5 could be removed since they present common knowledge.

Figure 6,7 and 8 are shown as examples but it would have been interesting to use them for comparison purposes between distraction and baseline (lane keeping)

Figure 12 should be a table as shown in table in discussion (there is no table with it).

There is no information regarding the accuracy with which participants answered the task. It is important for ensuring that they are involved in the task.

Conclusion is an unnecessary repetition of the abstract. It should focus on the advancement of knowledge of the current results and the steps to come.

Reviewer 2 Report

The authors presented the EEG-based driving distraction using 10 volunteers.

The idea is fine and unique. Nevertheless, the paper seems not well prepared.

1. The manuscript is not well formatted. The non-uniform or misformatted font (such as title), paragraph, and spacing were found.

2. The acknowledgment, authors' contribution, and conflict of interest statements are missing. The reviewer believes this complicated data collection should acknowledge to a particular group of respondents or community. Also, the financial support statement, is it by the funded body or by the authors?

3. Please do a short review and discussion regarding related studies of EEG for brain monitoring for the different case of studies. Do a comprehensive comparison in a table, and emphasize what the novelty is and the finding in this study.

4. Add photographs or documentation (of course, with non-disclosure privacy of the respondents) during the measurements to give the idea of how the data were taken. It can be uploaded as supplementary information.

5. The authors only provide the average ages of the respondents. This information is not enough. Please provide the standard deviation of the ages as well as gender distribution and occupation. Then it gives a better idea of observing the measurement of the respondents.

Major revision of the manuscript is strongly recommended.

Reviewer 3 Report

This manuscript presents an application of EEG in the classification of distration in driving. To achieve such a purpose, it uses brain connectivity estimators as features. The two experiments use a simple math problem solving (keepig the driving lane) and a non-distracted driving event (change lane)
In general, the work is interesting, but the manuscripts requires few improvments to allow a better novelty identification.
The authors are require to address few minor amendments, listed below.
A mathematical problem to solve a simple equation is recurrently cited allong the manuscript. It is interesting to presente the mathematical problem to allow the reader to infer about the simplicity of the problem. At the same time, it is interesting to present the mean time consumed by the 10 volunteers to solve this problem, in order to get an idea of how distracting is the task.
It is interesting to clarify the option for the selection of the GGC, PDC, GPDC and DTF brain connectivity estimators in the analysis.
Give more details about the instrumentation used in sections 2.1/2.3 (initial stage) and a photograph.
Figure 1 is not a block diagram, but a preview/graphical description of the approch followed.
In section 2.2, on line 131, it is important to give more details about the dynamic motion simulator. Is it an arcade set with moving chair, a computer based program, a XBox/PlayStation console or a virtual reality environment. Which was the platform used to mimitize the driving experience? Please clarify.
In section 2.3, on line 162, please provide more details about the low-pass filter, e.g., the order, the maximum attenuation in the pass-band, the minimum attenuation in the stop-band, the size of the transition-band.
Figure 4 is not also a block diagram, but the sequence of stepps followed during thr brain connectivity analysis.
In Figure 5, is there any special reason to select a time duration of of 50ms for the time segments?
What was the criterion to select the ICA outputs in Figure 6 for a specific participant?
The filtered ICA outputs on Figure 7 reffers to the same participant of Figure 6?
WHich the ICA ouputs of Figure 7 correspond to the selected ICA outputs in Figure 8?
In page 343, it was interesting to better explain the application of the ficsvm function in the context of the signals obtained from the different volunteers
In page 344, why 30 objects?
Please make a better explanation of how the results in figures 14 and 15 with the connectivities were obtained in the scope of the 10 volunteers.
On line 421, these ICs have any relations with the ones in figure 8 or with something else?
What was made of IC5 and IC6 from Figure 8 in the scope of the connectivities? These ICs were discarded for any reason?
On line 447, is there any possibility of this being implemented automatically and thus, to result on a commercial product to measure the competence of drivers as part of psicotechnical tests?

Round 2

Reviewer 1 Report

there is repetition between paragraph at line 57 and 61, please review the text so it is more fluid.

line 64, in the study (25) should be reviewed because the presentation of this work is inappropriate based on the format of the reference.

line 88, student is there twice

paragraph at line 104 would be better placed in the discussion closer to the conclusion

quality of figure 1 as to be improved (resolution)

line 145 a real car (s real car)

line 145: is it VR os simulation, they are different technologies and should be named accordingly, the one presented here looks like simulation.

As the speed being fixed at 100kmh ? If so, it should be noted as a limitation since it does reduce the cognitive cost of driving.

line 163 has to be reviewed for clarity. The same sentence is repeated on line 187.

ICs are presented differently in figure 5 and 6, please modify accordingly

As stated in Figure 7, why is figure 5 not the same participant as fig 6 and 7 as well.

line 325, label of lobe and cortex are incomplete

line 340 does not match with table 1 (HQ is 9 instead of 8 in the table)

paragraph line 336 is all cited in paragraph line 343, please revise

What is "cognitive power"? It has not been described anywhere. Do you mean cognitive capacity?

Line 420 and Fig 12 should go in results

line 473 "simulated" driving experiment 

an opening should be made at the ned of the conclusion instead of preparing the results obtained.

Nowhere is there a statement regarding an ethics review board or a formal consent form. This is of utmost importance and should be mentioned where and when it was signed by participants. I am concerned it might not have been approved, and therefore, I would like to see the reference number of it and the home institution that approved it.

With all these points, the paper is still not suitable for publication.

Reviewer 2 Report

The manuscript has been revised extensively by the authors, and the scientific contents are presented clearly than the first version.

The reviewer recommends the manuscript for publication in Sensors.

Author Response

The authors would like to thank Reviewer2 for recommending the manuscript for publication in Sensors.